# Production of Aflatoxin B1 by *Aspergillus parasiticus* Grown on a Novel Meat-Based Media

**DOI:** 10.3390/toxins15010025

**Published:** 2022-12-29

**Authors:** Iva Zahija, Barbka Jeršek, Lea Demšar, Mateja Lušnic Polak, Tomaž Polak

**Affiliations:** Department of Food Science and Technology, Biotechnical Faculty, University of Ljubljana, Jamnikarjeva 101, 1000 Ljubljana, Slovenia

**Keywords:** moulds, *Aspergillus parasiticus*, mycotoxins, aflatoxin B1, meat-based media

## Abstract

The aim of the present study was to develop meat-based media with compositions similar to those of dry-fermented meat products and to evaluate their use in studying the growth of *Aspergillus parasiticus* and the kinetics of aflatoxin B1 (AFB1) production. In our previous experiments, we found that the strain *A. parasiticus* ŽMJ7 produced a high amount of AFB1. Cooked meat agar (CMA2) was used as a novel complex meat-based medium with four variations: CMA2G (CMA2 supplemented with 1% glucose), CMA2YE (CMA2 supplemented with 0.2% yeast extract), and CMA2GYE (CMA2 supplemented with 1% glucose and 0.2% yeast extract). Media were inoculated with an *A. parasiticus* spore suspension (10^5^ spores/mL) and incubated at 25 °C for up to 15 days. The *A. parasiticus* lag phase lasted less than 1 day, irrespective of the growth medium, with the exception of control medium CMA1 (cooked meat agar) as an already known meat-based medium. The highest mean colony growth rate was observed on CMA2 and CMA2G. Reversed-phase UPLC–MS/MS analysis was performed to determine the AFB1 concentration in combination with solid phase extraction (SPE). The highest AFB1 concentration in meat-based media was detected in CMA2GYE after 15 days of incubation (13,502 ± 2367 ng/mL media). The results showed that for studying AFB1 production in dry-fermented meat products, novel suitable media such as CMA2-based media are required. This finding could represent a potential concern with regard to the production of dry-fermented meat products.

## 1. Introduction

Moulds are a natural part of the environment, and they have the ability to adapt and colonise various substrates. Mould contamination is a challenge for food safety given its undeniable role in spoilage and the possible toxic impact on human health and the economy. Mycotoxins are secondary metabolites of fungi and are typically produced by *Aspergillus*, *Penicillium*, *Alternaria*, and *Fusarium* strains [1,2]. The biological functions of mycotoxins are related to the adaptation of fungi to the survival environment (food medium) [3]. Mycotoxin biosynthesis depends on fungal species, substrate composition, carbon and nitrogen source, temperature and water activity of the media, pH, relative humidity, bioactive agents, and storage or incubation time [4,5]. These factors do not act independently, and complex interactions occur between them [6,7,8]. Among mycotoxins, aflatoxins (AFs) are highly carcinogenic, which is particularly relevant in the case of human consumption of contaminated food [9,10]. AFs are produced primarily by *Aspergillus parasiticus* and *Aspergillus flavus* strains. Toxigenic *A. flavus* strains produce AFB1 and AFB2, whereas *A. parasiticus* strains produce AFB1, AFB2, AFG1, and AFG2. The toxicity levels of AFs exhibit the following order: AFB1 > AFG1 > AFB2 > AFG2 [11]. Among them, AFB1 has been identified as a proven human carcinogen and is classified as a Group 1 carcinogen [12]. AFB1 is one of the most potent naturally occurring carcinogens and is the most toxic member of this group of mycotoxins. In foods of plant origins, primary contamination with AFB1 occurs when aflatoxin-producing moulds colonise a foodstuff under favourable conditions (suitable relative humidity, temperature, aeration, the presence of insects, and physical damage to the substrate). Unlike foods of a plant origin, contamination of foods of an animal origin may be classified as primary (production of AFs by moulds present during ripening) or secondary (the carry-over effect). The latter implies the transfer of AFs from animal feed via animal to food [13], as well as indirect contamination of products with additives and ingredients used in meat product technologies [14].

Dry-cured fermented meat products generally have low water activity (a_w_), high salt content, neutral to low pH, and varying ripening times (from a few weeks to a few months) at different temperatures. These conditions favour xerophilic and xerotolerant mould growth, such as the *Aspergillus*, *Eurotium,* and *Penicillium* species [15]. In some dry-cured meat products, such as sausages, salami, and hams, surface mould colonisation is important because of their lypolitic and proteolytic activities, which lead to meat product-specific aromas, flavours, and textures. When a starter mould culture is not used or when mould growth is not controlled during ripening, some undesired moulds may also grow, and could be responsible for spoilage or even mycotoxin formation [16]. Ochratoxin A (OTA) is classified by the IARC as a possible human carcinogen (Group 2B), and is the dominant mycotoxin contaminant in meat products [17]. OTA is the mycotoxin most commonly found in dry-cured fermented products [18,19,20,21]. OTA has been found in meat products produced from the meat of animals fed with OTA-contaminated feed [22], on the surface of dry-cured Iberian ham [23], in dry-cured meat products [24], in Italian salamis [25], in Croatian Kulen [14,26], in the casing of dry fermented sausages produced in Brazil [27], and in Egyptian beef sausages [28].

AFB1 occurs less frequently in meat products and in lower concentrations than OTA [14,20,21,29,30], although AFs-producing moulds *A. flavus* and *A. parasiticus* were isolated from dry-cured meat products [31,32]. AFB1 was found in Croatian fermented meat products [20] and in Croatian Kulen [26].

To date, most studies have focused on the influence of ecophysiological conditions (a_w_, temperature, and salt and glucose content) on the growth and production of mycotoxins on different culture media [33,34,35,36]. Several studies have assessed the effects of abiotic parameters, salt, and spices on OTA production by *Penicillium* strains in meat-based model systems (media containing some meat components), primarily including dry-cured meat media [37,38,39,40].

Only a few studies have focused on the effects of environmental conditions (processing and ripening conditions) and substrate on aflatoxigenic *Aspergillus* strains [24,41,42,43] regarding their growth and AFB1 production. These studies used different meat-based model systems that enable the assessment of optimal and marginal conditions for mould growth and AF production. The main advantage of such studies is that they allow the assessment of different environmental parameters, various species of fungi, and their interactions both in a controlled environment and in meat product-like media. *A. flavus* and *A. parasiticus* are closely related species that produce aflatoxins, but these physiological properties are strain-related, although the majority of strains are capable of synthesizing aflatoxins [44,45,46]. In our previous experiments (data not included in the manuscript), we found out that the strain *A. parasiticus* ŽMJ7 produced the highest amount of AFB1 (24.887 ± 4.456 mg/L on YES), which was 5–10 times more than other tested strains of *A. flavus* and *A. parasiticus*. The high AFB1 productivity was the main reason for using this strain in the present study. Additionally, recent research has shown that *A. parasiticus* strains have great potential to contaminate foods and produce AFB1 [47,48]. Given that the substrate is one of the most important factors controlling the production of secondary metabolites [49], the aim of this study was (i) to design inexpensive meat-based media that will simulate the composition of dry-fermented meat products; (ii) to evaluate its usage for studying the growth of *A. parasiticus* and AFB1 production kinetics; and (iii) to assess suitability of the media for AFB1 production studies of *A. parasiticus*.

## 2. Results and Discussion

### 2.1. A. parasiticus Growth Assessment

*A. parasiticus* colonies were observed on all media after 24 h of incubation at 25 °C, irrespective of the type of medium (Figure 1 and Figure 2). Within the following days of incubation, the highest mean colony diameter was obtained on YES. Colonies of *A. parasiticus* attained a 70- to 75-mm diameter after 7 days of growth at 25 °C on YES, CMA2, and CMA2G. However, on MEA, CYA, CMA1, CMA2YE, and CMA2GYE, the colonies grew to reach 50–65 mm in diameter in the same period (Figure 1 and Appendix A).

Colony colour after 7 days varied from green on YES, MEA, CYA, and CMA2G to white and pale brown/orange on CMA1, CMA2, CMA2YE, and CMA2GYE. Colony colour after 14 days varied from distinctive green on YES, MEA, CYA, CMA2G, and CMA2GYE to pale brown/orange on CMA1, CMA2, and CMA2YE. The addition of glucose to CMA2 led to green colony colour formation on CMA2G and CMA2GYE, whereas the addition of yeast extract to CMA2 led to brown colony formation on CMA2YE (Appendix A and Figure 2). The substrate and the addition of glucose and yeast extract affected the colony colour on the media used (Figure 2). The effect of the incubation period and the substrate on colony colour is shown in Appendix A.

### 2.2. A. parasiticus AFB1 Production Analysis

AFB1 analysis revealed that the growth and production of secondary metabolites of the *A. parasiticus* strain were affected by the medium (substrate) and incubation period, at 25 °C, under optimal growth temperatures, as reported by Peromingo et al. [41] for *A. parasiticus*. Figure 3 summarises the effect of the different media on AFB1 production kinetics by *A. parasiticus*. Yogendrarajah et al. [50] noted that it is essential to report the toxigenicity of the fungal strain together with the growth substrate and growth conditions. AFB1 was detected in all tested media in which mould growth was observed. In general, the highest yield of AFB1 was produced on YES after 12 days of incubation (27,645.57 ± 2401.50 ng/mL media) (Appendix A). This result is in agreement with those reported by Lozano-Ojalvo et al. [34], who demonstrated that high AFB1 production was observed by *A. parasiticus* on a yeast extract sucrose peptone medium at the end of the incubation period, at 96 h. Razzaghi-Abyaneh et al. [51] reported 73.2-fold higher AFB1 production by *A. parasiticus* after 96 h of incubation in YES broth. The production of AFB1 by *A. parasiticus* was two-fold higher in comparison with the study by Yahyaraeyat et al. [52]. Although *A. flavus* is considered as the most predominant aflatoxin producer, recent research has shown that *A. parasiticus* strains have great potential to contaminate foods and produce AFB1 [47,48]; hence, this was the additional rationale for using *A. parasiticus* over *A. flavus* in the present study.

The highest AFB1 levels were found on the YES medium on all days of incubation; in contrast, the lowest AFB1 levels were found on the CMA2YE medium (Appendix A; Figure 3). On the first day of incubation, 18.36 ± 7.05 ng AFB1/mL was produced on YES (Appendix A). Armando et al. [53] reported that 347.5 ± 0.28 ng AFB1/mL was produced by *A. parasiticus* after a 24-h incubation period on YES. This difference in AFB1 value might be due to different *A. parasiticus* strains. Yogendrarajah et al. [50] reported that isolates of *A. parasiticus* grown on MEA at 22 °C produced 35–12,000 μg AFB1/kg. However, at 30 °C, the production of AFB1 was less than 1–6500 μg kg^−1^. Our *A. parasiticus* strain produced a similar amount of AFB1 (24–3900 ng/mL) when grown on MEA at 25 °C.

Peromingo et al. [41] reported that the optimum growth parameters for *A. parasiticus* were 25 °C and 0.95 a_w_ on meat-based and dry-cured ham-based media.

The highest AFB1 level on meat-based media was detected on CMA2GYE on the last day (15) of the incubation period (13,502.39 ± 2367.68 ng/mL media). Glucose had a stimulating effect on AFB1 production by the *A. parasiticus* strain. When 1% glucose was added to CMA2, AFB1 production increased 13.9-fold on CMA2G on the last day of incubation. Sugars (glucose, saccharose, and sometimes lactose) are commonly used in the industrial manufacturing of fermented meat products, although traditional processes do not commonly use this ingredient. During fermentation and ripening, lactic acid bacteria convert glucose (their primary energy source) to lactic acid, which is the main component responsible for the pH decrease. This acidification has a preservative effect, dominating the competitive microflora, and contributes to the development of the typical sensory characteristics of the fermented sausages [54]. Sugar addition is a well-established practice in the meat industry to facilitate the fermentation process and ensure safety [55]; addition of 1% glucose reduces the pH by 1 unit [56]. In contrast, yeast extract had an opposite effect on the capacity of AFB1 production by the strain. When 0.2% yeast extract was added to CMA2, AFB1 production decreased 13.2-fold on CMA2YE on the last day of incubation. Yeast extract is a natural ingredient that is extensively used in the food industry to improve flavour, and contains a rich source of compounds and precursors, such as peptides, nucleotides, group B vitamins, and amino acids. Yeast extract can be applied in food formulations in low amounts (usually less than 1%) [57,58]. Yeast extract has been widely used as a flavouring agent and as a precursor to the formation of compounds that have pleasurable tastes and aromas in meat products [59]. When glucose and yeast extract were added simultaneously to CMA2, a stimulating effect on AFB1 production was detected. On the other hand, sugar and yeast extracts are common ingredients in various microbial culture media (YES, CYA). AFB1 levels were rather high on YES and CYA on the last day of incubation (Appendix A). It seems that glucose and yeast extract had a synergistic effect on AFB1 biosynthesis in these media. The level of AFB1 on CMA2GYE was 4.3-fold higher than that on CMA2G, 60-fold higher than that on CMA2, and 795-fold higher than that on CMA2YE on the last day of incubation (Appendix A). Thus, in the presence of glucose or glucose and yeast extract, the nutrients available in meat-based media (CMA2) support AFB1 biosynthesis by *A. parasiticus*. Media with the lowest levels of AFB1 on the last day of incubation were CMA1, CMA2, and CMA2YE, with AFB1 levels lower than 230 ng/mL (Appendix A), whereas green colouration was not observed only on these media (Appendix A). The AFB1 concentrations on CMA1, CMA2, and CMA2YE were not significantly different on the last day of incubation (Appendix A).

AFB1 production kinetics was different on meat-based media simulating the composition of dry-fermented meat products (CMA2GYE) and microbial culture media (YES and MEA) (Table 1). Thus, it seems essential to evaluate food-borne mycotoxigenic fungi on food-based substrates.

On the other hand, the AFB1 production rate was similar on CMA1 and CMA2 (Table 1), although their composition was not alike (CMA1 has beef heart extract, glucose, meat peptone, and sodium chloride in its composition). However, when glucose was added to CMA2 (CMA2G), the AFB1 production rate was significantly different than on CMA1 (Table 1). Thus, the usage of CMA1 as already known meat-based media could be a good basis for study of toxigenic fungi isolated from a meat-based environment.

The usage of CMA2GYE, as a novel complex inexpensive meat-based medium, seems more suitable for an AFB1 production kinetics study in comparison with other meat-based media used in this experiment (CMA1, CMA2G, and CMA2YE) (Figure 3).

To our knowledge, this is the first study that has analysed the AFB1 production kinetics of aflatoxigenic fungi (*A. parasiticus*) on three different microbial culture media (YES, MEA, and CYA) and five meat-based media (CMA1, CMA2, CMA2G, CMA2YE, and CMA2GYE), including observations of colony growth and quantification of AFB1 every day during a 15-day incubation period at 25 °C. Previous studies have shown the toxigenic potential of *A. parasiticus* on various microbial culture media; however, AFB1 determination was not performed every day during the incubation period [34,42,50]. Although *A. flavus* is the most predominant aflatoxin producer, our results showed that *A. parasiticus* should also be considered as a potent threat to meat technologies and, consequently, to human health.

### 2.3. Relationship between Growth and AFB1 Production

Regarding growth assessment, the effect of different media on the lag phase (λ) or the time to visible growth and growth rate (μ) of *A. parasiticus* is shown in Table 1. The *A. parasiticus* lag phase lasted less than 1 day, irrespective of the growth medium, with the exception of CMA1 (1.02 ± 0.04). Lag phases were significantly shorter on YES, MEA, and CMA2YE compared with other media. The distributions of the medians for lag phases between the different media were significantly different (*p* < 0.05). The values for the lag phase on CYA, CMA2, CMA2G, and CMA2GYE media are consistent with those observed by Sánchez-Montero et al. [43], who reported an *A. parasiticus* lag phase of 0.81 ± 0.21 days on a dry-cured meat model system. Data plots showed, after a short lag phase, a linear trend of growth with time on all media until reaching a stationary phase or a plateau. The growth of *A. parasiticus* colonies was limited by the size of the medium in the Petri dish (YES, CMA2, and CMA2GYE) and/or by metabolic product(s) that may inhibit colony growth. The mean colony diameter of *A. parasiticus* was the largest on CMA2G on Day 7 (73.2 mm), 31.3% greater than that on CMA1 (50.3 mm), which yielded the smallest colony diameter. The highest mean colony growth rate (μ) was observed on CMA2 (10.38 ± 0.29 mm/day), CMA2G, and YES, followed by CMA2YE, CMA2GYE, CYA, and CMA1, and the slowest was observed on MEA (5.42 ± 0.42 mm/day). The growth rates on meat-based media were greater than those obtained by Sánchez-Montero et al. [43]. This finding may be because different meat-based media were used in this study.

*A. parasiticus* strains grown on MEA at 22 °C had a lag phase of 0.54–1.25 days, and reached a maximum radial growth rate of 2.10–5.57 mm/day. At 30 °C, these strains reached a lag phase of 0.85–0.90 days and maximum radial growth rates of 5.15–5.90 mm/day [50], which was consistent with our results.

The change in colony colour (formation of green colouration) on CMA2GYE, starting from Day 7 (Figure 1 and Appendix A), is associated with mould maturation, which might be associated with a higher rate of AFB1 production (k AFB1). Significantly higher k AFB1 on CMA2GYE was determined from Days 13–15 (2147.63 ± 780.32 ng/mL/day) in comparison with values obtained from Days 2 to 11 (629.07 ± 36.86 ng/mL/day). The growth of *A. parasiticus* colonies was limited by the size of the medium in the Petri dish on CMA2GYE on Day 15 (Figure 3), whereas fungi started to produce AFB1 at a higher rate approaching the end of the incubation period. The rate of AFB1 production from Days 13–15 on CMA2GYE was not significantly different from that on YES (2895.00 ± 368.42 ng/mL/day), which had the highest k AFB1 in the experiment under the optimal growth conditions used in this study. On CMA2YE, the lowest k AFB1 was observed (1.76 ± 0.31 ng/mL/day). The distributions of the medians of AFB1 production rates (k AFB1) between the different media were significantly different (*p* ≤ 0.05). On the last day of incubation, the level of AFB1 on YES was approximately 1440-fold higher than that on CMA2YE. The main finding here is that although no progressive visual growth of fungi occurred at the end of the incubation period (due to limited space in Petri dishes or by and metabolic product(s) that inhibited colony growth), this does not necessarily mean that mycotoxin production is not in progress. AFB1 production rates for CMA2 and CMA2GYE were also determined for period between 13 and 15 days, as a significant rise in AFB1 was noticed in the results in comparison to Days 2–11. AFB1 production rates determined for CMA2 and CMA2GYE from Days 13 to 15 were significantly higher than those from Days 2 to 11 (*p* ≤ 0.05). However, in the same period, fungal growth entered the stationary phase. Our results indicate that the effect on growth and AFB1 production depended on the media considered and the incubation period, and this finding is consistent with that of Gil-Serna et al. [60].

CMA2GYE was the second medium in this study with the highest yield of AFB1, which is concerning from the view of the production of dry-fermented meat products (DFMPs). The k AFB1 for CMA2GYE was significantly different for the periods from 13 to 15 days and from 2 to 11 days (*p* < 0.05). A two-week production time for fast fermented meat products could theoretically be long enough for high AFB1 accumulation, if aflatoxigenic fungi were present. Components intended to simulate the additives used in the industrial production of salami/dry-fermented meat products stimulated AFB1 formation on CMA2GYE meat-based media containing glucose and yeast extract. This finding is concerning from the food safety point, as it is of crucial importance that the meat industry controls hygiene in ripening chambers. It was pointed out that air inside them may be an important source of contamination [61] to reduce the possibility of mould contamination, and, consequently, the possibility of AFB1 accumulation in their products. On the other hand, uncontrolled environmental conditions in rural households could cause a great risk for AFB1 contamination in meat products. The combination of environmental conditions favourable for AFB1 production may be temporary conditions during processing and storage, but the generated contamination could last until the final step of production. This is due to AFs being chemically stable during storage and processing, and being relatively heat-stable within the conventional food processing temperature range. It is important to avoid the conditions that lead to their formation [62]. In our research, the main purpose was to study AFB1, as the most potent AF carcinogen. It would certainly be very important to also assess the formation of AFB2, AFGI, and AFG2.

The MEA and CYA media supported fungal growth (Figure 1), but not efficient AFB1 production (Figure 3). The levels of AFB1 on MEA and CYA were approximately 4.5- to 6.7-fold lower than those on YES on the last day of incubation. On the other hand, YES supported efficient *A. parasiticus* growth and AFB1 production. Thus, mould growth as the only parameter should not be considered a good predictor of AF contamination. A similar phenomenon regarding mycotoxin production and fungal growth on MEA was previously demonstrated by Muñoz et al. [35], as the MEA medium supported fungal growth, but not efficient OTA production by *P. nordicum,* on coffee- and wheat-based media.

A slight reduction in AFB1 levels was observed at day 12 for YES and CYA, which was statistically significantly different (*p* ≤ 0.05). This finding is indicative of some degradation in the colony. A decrease in the rate of AFB1 accumulation occurred on YES once a slight decrease in growth rate was observed (Figure 1), based on colony diameter, due to the limited space in the Petri dish. A decrease in the rate of AFB1 accumulation occurred on CYA, but without a decrease in the growth rate. Garcia et al. [63] concluded that the decrease in the rate of toxin accumulation by *A. flavus* in maize-based media might be due to (i) a decrease in the growth-associated toxin accumulation due to the decrease in the growth rates; (ii) a decrease in the rate of toxin production by the existing biomass; or (iii) a degradation of toxin by the existing biomass. Although we used different media and moulds, the same phenomenon regarding the decrease in the rate of toxin accumulation occurred in YES and CYA. 

AFB1 production kinetics on YES and MEA were significantly different than those on all meat-based media in this study (Table 1), so usage of meat-based media with undesirable mould strains that could be responsible for spoilage or mycotoxin formation could be a good strategy for ensuring food safety.

Garcia et al. [63] reported that aflatoxin formation by *A. flavus* in maize-based media resulted from a combination of growth rate and cell concentration, and occurred during both growth and stationary phases (a mixed-growth associated trend). In our experiment, AFB1 production showed a similar mixed growth-associated trend as that described by Garcia et al. [63], which occurred during growth, suggesting a behaviour similar to that of primary metabolites and the stationary phase (Figure 3) in all media. This finding implied that the toxin is either a metabolite produced by growing cells or is converted biosynthetically from some other compound by growing cells. We can assume that the *A. parasiticus* AFB1 production kinetics may follow a mixed growth-associated trend under the given conditions and media used in the experiment. One may expect the production of mycotoxins as secondary metabolites to follow a curve parallel to that of growth, but slightly delayed. However, regulation of the secondary metabolism is poorly understood [64], and the relationship between the rates of primary and secondary metabolism is unclear [65]. 

In a study by Lozano-Ojalvo et al. [34], lower AFB1 values were obtained when *A. parasiticus* was grown on a substrate containing sodium nitrate. This finding is similar to our findings; the CYA medium contained lower AFB1 values than YES, which lacks sodium nitrate, throughout the whole incubation period (Appendix A). It was previously reported that sodium nitrate-containing media do not support aflatoxin production [66]. On the other hand, nitrate and nitrite are used to cure meat products. Nitrate and nitrites are mostly added as potassium or sodium salts, and their usage is regulated by law. Nitrite acts primarily as an inhibitor for some microorganisms, and has an important role in forming a typical cured meat colour [67]. From our results, it can be concluded that sodium nitrate in CYA seems to inhibit AFB1 biosynthesis to some extent.

CMA2, CMA2G, and CMA2YE had rather low R^2^ DIAM/AFB1 values (R^2^ < 0.8). These findings indicate that fungi grow rapidly, but the production of AFB1 did not follow. MEA, CYA, and CMA1 had rather high R^2^ DIAM/AFB1 values (R^2^ > 0.9), indicating that fungi had moderate μ and moderate k AFB1 values. YES and CMA2GYE had rather high R^2^ DIAM/AFB1 values (R^2^ > 0.9), indicating that fungi grow rapidly and have rather high k AFB1 values (Table 1). This parameter (R^2^ DIAM/AFB1) also confirms that mould growth should not be the only parameter considered a good predictor for AF contamination.

There are more data regarding OTA biosynthesis by *A. flavus* in dry-cured meat products and meat-based media [38,42,43] than that for AFB1 production kinetics in correlation with *A. parasiticus* and meat-based media. Therefore, we believe that with our study, an information gap about AFB1 formation on media containing meat components can be filled. During the processing steps, especially ripening, the presence of moulds on the surface of DFMPs is visible in a wide range of environmental conditions [18,20,68]. However, some moulds are objectionable based on their toxigenic potential [23,24,30]. The presence of visible moulding is even required during the ripening step to preserve the tradition in many typical products and, perhaps, to achieve the development of classic flavour and colour.

## 3. Conclusions

Substrate (medium) composition has a significant influence on the toxigenic potential of the same fungal strain under the same growth conditions. *A. parasiticus* required a very short time to adapt, colonise, and start producing AFB1 on meat-based media which contained components intended to simulate the additives used in the production of salami/dry-fermented meat products. This occurred within the 15-day incubation period at 25 °C.

Based on these results, mould growth should not be the only parameter considered as a good predictor for AF contamination and toxigenic moulds isolated from food-based media should be tested on food-based medium as AFB1 production kinetics showed different play on meat-based media in comparison with microbial culture media.

The results obtained in this work suggest that *A. parasiticus* should be considered a great potential risk for AFB1 contamination due to its high capacity for producing AFB1 in meat-based media.

Based on these results, uncontrolled mould growth during the early phases of production (when temperatures in the ripening chambers are rather high) must be effectively controlled, as it can lead to mycotoxin contamination of dry-fermented meat products. Even low levels of AFB1 produced in DFMPs, as shown with the meat-based media in the present study, may contribute to daily consumption, since DFMPs are a significant component of the Mediterranean diet.

Although more studies are needed to simulate environmental conditions during ripening with a combination of various fungi and supplemented meat-based media, our results suggest that temporary conditions during processing and storage could generate rather high levels of AFB1 contamination of DFMPs, which is an important food safety hazard. This finding may highlight a problem for the industry, given that no AFB1 controls are currently necessary for DFMPs. Our developed inexpensive meat-based medium is a good basis for further research, as it offers an easy method of preparation and numerous possibilities for supplementation with different compounds.

Further research should be focused on the longer incubation period (simulating more production parameters of dry-fermented meat products), using larger medium/Petri dish diameters and supplementing CMA1 with various additives that can simulate meat-based matrices in more detail.

## 4. Materials and Methods

### 4.1. Fungal Strain and Inoculum Preparation

*Aspergillus parasiticus* ŽMJ7 strain (ŽMJ is designation of the Culture Collection of Laboratory of Food Microbiology at Dept. of Food Science, Biotechnical Faculty at the University of Ljubljana, Slovenia) was used in this study. *A. parasiticus* was isolated from the ripening chamber in traditional salami production in Savinjska Valley, Slovenia. Identification of the strain was performed according to morphological characteristics and confirmed by sequencing the β-tubulin gene [42]. *A. parasiticus* ŽMJ7 was selected according to previous results (not shown) of testing different strains for AFB1 production, as it produced the highest content of AFB1 when grown on YES (yeast extract sucrose agar) at 25 °C for 7 days (24.887 ± 4.456 mg/L medium). *A. parasiticus* was grown on malt extract agar (MEA, Sigma—Aldrich Chemie GmbH, Steinheim, Germany) at 25 °C for 7 days. Spores were collected with a sterile inoculation loop and suspended in sterile 0.5% agar with Tween 80 (0.1% *v*/*v*). The spores were quantified using a Thoma counting chamber (Brand, Wertheim, Germany), and the spore suspension was adjusted to 10^5^ spores/mL and maintained at −80 °C. The *A. parasiticus* spore suspension was inoculated (10 µL) on MEA and incubated at 25 °C for 7 days, and a new spore suspension was prepared prior to each experiment [69].

### 4.2. Media and Experimental Design

In our experimental work, 2 meat-based media were used: CMA1 (Cooked Meat Agar 1) and CMA2 as a novel meat-based medium (Cooked Meat Agar 2 with 4 variations). In addition, 3 microbial culture media which are regularly used for mould growth were also utilized

(YES, MEA, and CYA (Czapek Yeast Agar)).
CMA1 was prepared using CMB (Cooked Meat Broth, Merck, Rahway, NJ, USA) with the addition of bacteriological agar (15 g/L);CMA2 was prepared as follows: veal bones (4 kg) were purchased from a local market, boiled for 4 h in 4 L of distilled water, and concentrated to 6% dry weight during cooking. The main advantage of this meat-based medium is its ease of preparation, as well as its numerous possibilities for supplementation with different compounds used in meat technologies. The resulting mixture was filtered through a double layer of muslin, and 20 g/L bacteriological agar was added prior to sterilization. The composition of CMA2 was chemically determined (water 94%, ash 1%, fats 0.7%, proteins 4.3%).CMA2G: CMA2 medium was supplemented with 1% glucose (D-(+)-glucose, Sigma–Aldrich Chemie GmbH).CMA2YE: CMA2 medium was supplemented with 0.2% yeast extract (Sigma–Aldrich Chemie GmbH).CMA2GYE: CMA2 medium was supplemented with 1% glucose and 0.2% yeast extract, as these components can be used in the standard production of dry-fermented meat products (and are elements of various microbial culture media).YES (20 g/L yeast extract, 150 g/L sucrose, 20 g/L agar, 1 g/L MgSO_4_ · 7H_2_O);MEA (30 g/L malt extract, 5 g/L mycological peptone, 15 g/L agar);CYA (5 g/L yeast extract, 30 g/L saccharose, 15 g/L agar, 10 mL/L concentrated Czapek, 1 g/L K_2_HPO_4_).

All media were autoclaved at 121 °C (103 kPa) for 20 min, then cooled down to 45–50 °C and vigorously shaken prior to pouring (25 mL) into 90-mm diameter Petri dishes.

To evaluate the influence of medium composition on *A. parasiticus* growth and AFB1 production, YES, MEA, and CYA were used as general microbial culture media. CMA1 was used as an already known meat-based medium, and CMA2, CMA2G, CMA2YE, and CMA2GYE were used as the novel complex meat-based media designed in the study. Media were centrally inoculated with 10 μL of *A. parasiticus* spore suspension (10^5^ spores/mL), and then incubated in the dark at 25 °C for up to 15 days. During the incubation period, *A. parasiticus* growth was measured, and samples for AFB1 analysis were obtained daily and frozen at −80 °C until further analysis (3 replicates per medium).

### 4.3. A. parasiticus Growth Assessment

*A. parasiticus* growth was determined daily by measuring the diameter of colonies in two directions at right angles to each other. All colonies were also photographed. Filamentous moulds grow on solid media, forming a circular colony around the initial inoculation point. Growth was measured until the colony reached the edge of the medium in a Petri dish (complete colonisation). These data were utilised for determination of the lag phase (λ) and the growth rate (μ). Data were analysed using a primary model by plotting colony diameter against time. The growth rate (μ (mm/day)) was obtained from the slope of the growth curve from days 2 to 7 (linear part of the slope). The lag phase (λ, days) was calculated from the linear part of the graph by equating the regression line formula to the size of the original inoculum [41,50].

### 4.4. AFB1 Analysis in Growth Media

#### 4.4.1. Purification of Samples

The content of AFB1 was determined using the modified extraction procedure described earlier [30] (Pleadin et al., 2015).

Every day, one Petri dish (from each studied medium) was removed from the incubator and stored at −20 °C prior to extraction and quantification. Briefly, for the determination of AFB1, the entire medium (25 mL) was extracted with 35 mL of 80% acetonitrile and 0.5% HCOOH in an ultrasonic bath (Branson 3510) for 15 min (Branson Ultrasonics Corporation, Brookfield, CT, USA). Afterwards, the mixture was passed through filter paper (Sartorius 388) and then hand-shaken. One millilitre (CMA1, CMA2, CMA2G, CMA2YE, CMA2GYE) and 0.250 mL (YES, MEA, CYA) of extract were evaporated to dryness under reduced pressure at 40 °C with a rotary evaporator (Buchi, R-215, Flawil, Switzerland), and later diluted with 3 mL of 10% acetonitrile. Extracts were cleaned using ISOLUTE Myco columns (60 mg/3 mL, Biotage, Uppsala, Sweden). After conditioning the columns with 2 mL of acetonitrile and 2 mL of 10 mM ammonium formate, 3 mL of a sample was applied to the columns. It was washed first with 3 mL of 10 mM ammonium formate and then with 3 mL of 10% acetonitrile. The columns were subsequently dried for 2 min under maximal vacuum and eluted with 2 mL of 0.5% HCOOH in acetonitrile. Samples were evaporated to dryness under reduced pressure at 40 °C with a rotary evaporator (Buchi, R-215). The obtained eluate was dissolved in 0.200 mL (CMA1, CMA2, CMA2G, CMA2YE, CMA2GYE) and 1 mL (YES, MEA, CYA) of 80% acetonitrile with 0.5% HCOOH. Afterwards, the samples were passed through a 0.2 μm nylon filter (Phenomenex, Torrance, CA, USA) and transferred into HPLC vials. Samples were analysed in duplicate for the presence of AFB1.

All chemicals and solvents used during the extraction process were of analytical grade. For the validation process, AFB1 standard CRM46304 (Sigma—Aldrich Chemie GmbH, Steinheim, Germany, LOT#XA25410V) was used.

Extracts were analysed by UPLC–MS/MS. AFB1 was quantified by the standard addition method, and the presence of AFB1 was qualitatively analysed.

#### 4.4.2. UPLC–MS/MS Analysis

To determine the concentration of AFB1, reversed-phase UPLC–MS/MS analysis was used for separation and quantification. The UPLC system used was the ACQUITY™ UPLC™ H-Class PLUS System (Waters, Milford, MA, USA) consisting of a thermostated sample manager FTN, a thermostatted column compartment, and a quaternary solvent manager. The UPLC system was coupled with a Xevo TQ-S micro triple quadrupole mass spectrometer (Waters). The chromatographic separation was performed using an ACQUITY UPLC^®^ BEH C18 column (1.7 μm, 50 mm × 2.1 mm). The conditions used were as follows: column temperature, 40 °C; injection volume, 2 µL; and flow rate of the mobile phase, 400 µL/min. The mobile phase components were 1 mM ammonium acetate in water + 0.5% acetic acid + 0.1% formic acid (*v*/*v*) (solution A) and methanol + 0.5% acetic acid + 0.1% formic acid (solution B) (720007377, September 2021, Waters). The mobile phase gradient was programmed as follows: 0.00–0.70 min, 5% B; 0.70–6.50 min, 5–50% B; 6.50–9.50 min, 50–100% B; 9.50–12.50 min, 100% B; 12.50–12.60 min, 100–5% B; and 12.60–14.00 min, 5% B.

The mass spectrometer was operated in positive ionisation mode (ESI+). The following operating conditions were employed: electrospray capillary voltage, 0.75 kV; cone voltage, 30 V; extractor voltage, 2 V; source temperature, 150 °C; desolvation temperature, 600 °C; cone gas flow rate, 50 L/h; desolvation gas flow rate, 1000 L/h; and collision energy, 20 eV. The MRM transition was 313 > 285 for quantification and 313 > 241 for confirmation. The data signals were acquired and processed on a PC running MassLynx software (V4.2 SCN1017; 2020, Waters). The identification of AFB1 was performed by comparing the retention time and mass spectrometric data, and was quantified according to peak areas against previously determined calibration curves. The method calibration parameters for different media are presented in Appendix A.

### 4.5. Statistical Analysis

Statistical analysis was performed using IBM SPSS Statistics for Windows v.22.0 (IBM Corporation, Armonk, NY, USA). Data on growth rates and aflatoxin B1 production were tested for normality using the Shapiro–Wilk test [41]. The normality test was unsuccessful for all datasets; thus, nonparametric data analysis was performed using the Kruskal–Wallis rank sum test. One-way ANOVA was used to analyse the effects of the medium and incubation periods separately on AFB1 production in fungi. Duncan’s multiple range test was used as a post hoc test to analyse the mean comparison. The statistical significance was set at *p* ≤ 0.05. Spearman’s correlation was calculated between the diameter and AFB1 production (procedure: correlate).

## Figures and Tables

**Figure 1 toxins-15-00025-f001:**
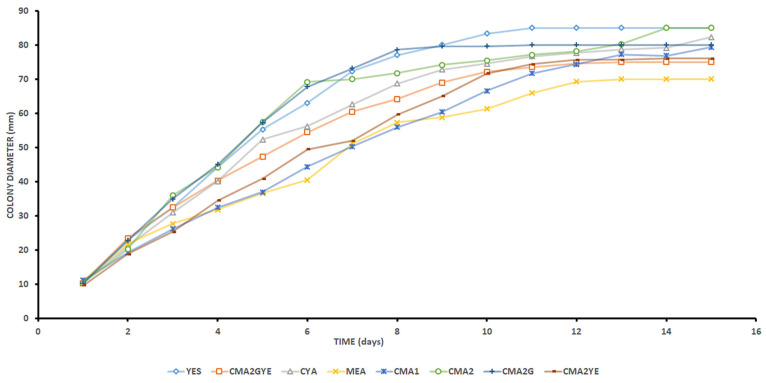
The mean colony diameter of *Aspergillus parasiticus* grown on YES (yeast extract sucrose agar), MEA (malt extract agar), CYA (Czapek yeast agar), CMA1 (cooked meat agar 1), CMA2 (cooked meat agar 2), CMA2G (CMA2 supplemented with 1% glucose), CMA2YE (CMA2 supplemented with 0.2% yeast extract), and CMA2GYE (CMA2 supplemented with 1% glucose and 0.2% yeast extract) during 15 days of incubation at 25 °C.

**Figure 2 toxins-15-00025-f002:**
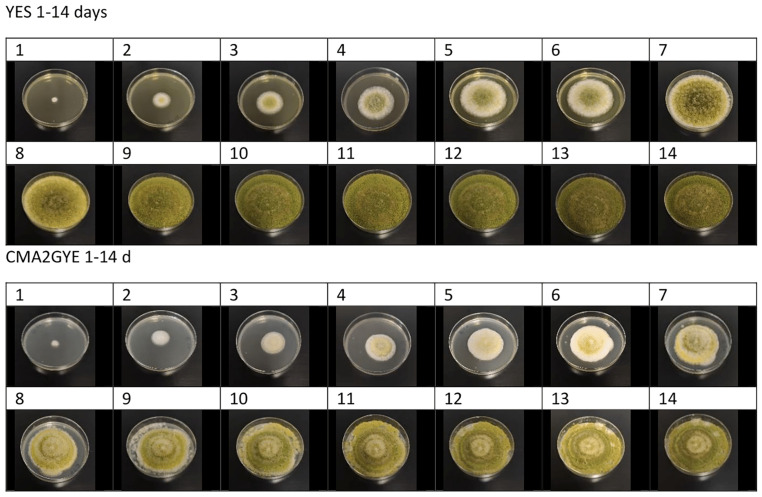
Examples of *A. parasiticus* colonies during 14 days of incubation at 25 °C on YES (yeast extract sucrose agar) and CMA2GYE (CMA2 supplemented with 1% glucose and 0.2% yeast extract).

**Figure 3 toxins-15-00025-f003:**
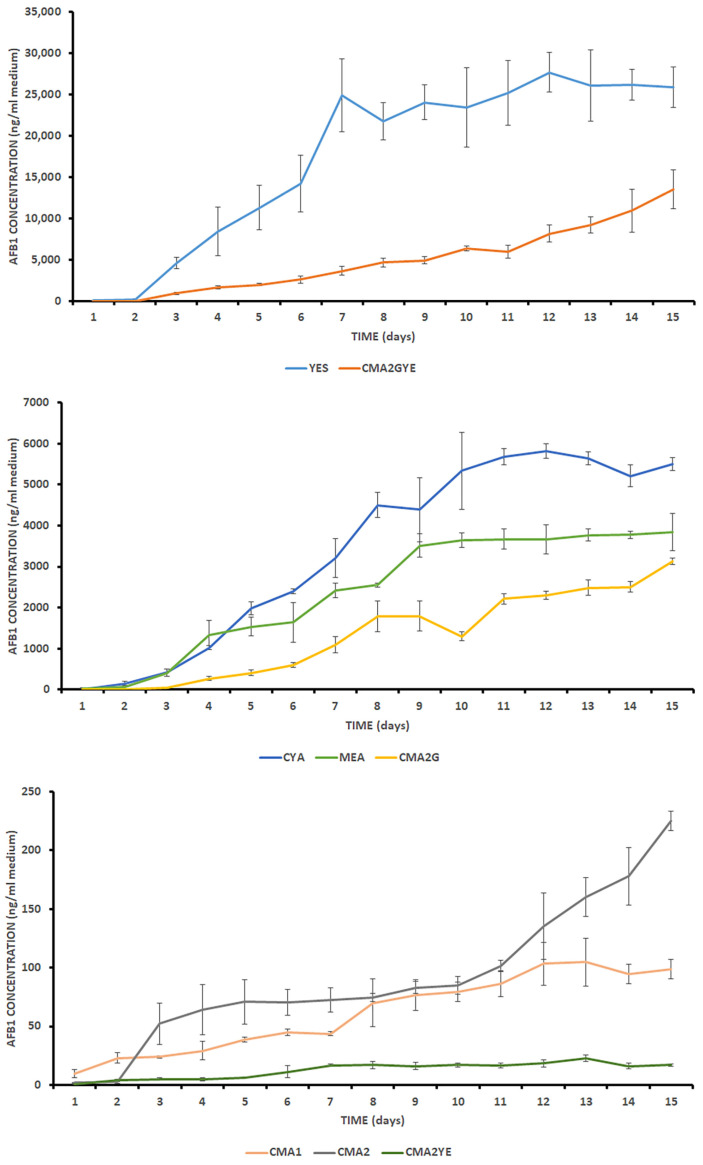
Aflatoxin B1 (AFB1) production of *A. parasiticus* grown on YES (yeast extract sucrose agar), MEA (malt extract agar), CYA (Czapek yeast agar), CMA1 (cooked meat agar 1), CMA2 (cooked meat agar 2), CMA2G (CMA2 supplemented with 1% glucose), CMA2YE (CMA2 supplemented with 0.2% yeast extract), and CMA2GYE (CMA2 supplemented with 1% glucose and 0.2% yeast extract) during 15 days of incubation at 25 °C.

**Table 1 toxins-15-00025-t001:** Average (mean ± SD) lag phase (λ), growth rate (μ), AFB1 production rate (k AFB1), and R^2^ (DIAM/AFB1) of *Aspergillus parasiticus* grown at 25 °C on different media (*n* = 3).

Medium	λ (Day)	μ (mm/Day)	k AFB1 (ng/mL/Day)	R^2^ (DIAM/AFB1)	Spearman
CMA1	1.02 ^a^ ± 0.04	6.11 ^bc^ ± 0.60	7.82 ^f^ ± 2.02	0.90 ^b^ ± 0.01	0.904
CMA2	0.82 ^b^ ± 0.10	10.38 ^a^ ± 0.29	7.43 ^f^ ± 1.55	0.63 ^c^ ± 0.10	0.928
CMA2*			32.39 ^e^ ± 12.37		
CMA2G	0.99 ^ab^ ± 0.16	10.35 ^a^ ± 0.08	248.49 ^d^ ± 23.08	0.79 ^d^ ± 0.01	0.924
CMA2YE	0.54 ^bc^ ± 0.55	7.64 ^b^ ± 0.57	1.76 ^g^ ± 0.31	0.77 ^c^ ± 0.10	0.795
CMA2GYE	0.67 ^b^ ± 0.60	7.37 ^b^ ± 0.07	629.07 ^b^ ± 36.86	0.93 ^a^ ± 0.01	0.979
CMA2GYE*			2147.63 ^a^ ± 780.32		
YES	0.45 ^c^ ± 0.18	10.03 ^a^ ± 1.30	2895.00 ^a^ ± 368.42	0.91 ^b^ ± 0.02	0.895
MEA	0.36 ^c^ ± 0.35	5.42 ^c^ ± 0.42	424.23 ^c^ ± 15.43	0.93 ^a^ ± 0.01	0.895
CYA	0.81 ^ab^ ± 0.62	8.43 ^b^ ± 0.59	663.30 ^b^ ± 68.17	0.93 ^a^ ± 0.05	0.912
Kruskal- Wallis test (*p*)	0.032	0.004	0.002	0.021	

Legend: YES (Yeast Extract Sucrose Agar), CYA (Czapek Yeast Agar), MEA (Malt Extract Agar), CMA1 (cooked meat agar 1), CMA2 (cooked meat agar 2), CMA2G (CMA2 supplemented with 1% glucose), CMA2YE (CMA2 supplemented with 0.2% yeast extract), and CMA2GYE (CMA2 supplemented with 1% glucose and 0.2% yeast extract). The lag phase (λ) was estimated by extrapolating the linear regression equation to the time axis. The growth rate (μ) was calculated from the slope of the growth curve from Days 2 to 7. The AFB1 production rate (k AFB1) was calculated from the slope of the AFB1 production curve from Day 2 to 11. The CMA2* and CMA2GYE* AFB1 production rates were also calculated from the slope of the AFB1 production curve from Days 13 to 15. The coefficient of determination (R^2^) between colony diameter (DIAM) and AFB1 formation in the exponential phase is shown. Spearman’s correlation was calculated between the diameter and AFB1 production. Different superscripts (a–g) along columns indicate significant differences (*p* < 0.05).

## Data Availability

Not applicable.

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
