# Peer review of "Production of Aflatoxin B1 by *Aspergillus parasiticus* Grown on a Novel Meat-Based Media"

_toxins, 2022, doi:10.3390/toxins15010025_

Round 1
Reviewer 1 Report
Dear Authors,
I have read very carefully the article " Production of aflatoxin B1 by Aspergillus parasiticus grown on a novel meat-based media " which I found in line with the aims of Toxin.
The aim of the current study was to (i) to design inexpensive meat-based media that will simulate the composition of dry-fermented meat products, (ii) evaluate its usage for studying the growth of A. parasiticus and AFB1 production kinetics and (iii) assess suitability of media for AFB1 production studies of A. parasiticus.
In my opinion, the manuscript is very interesting and the experimental design was well defined and discussed. I only have slight comments / suggestions.
The resolution of figure 1 and 2 must be improved. Concerning figure 1 it is difficult to follow and agree the comments reported in page 3 lines 98-102.
In table 2 and table S2 must be reported n, the number of independent variables you have. Please, carefully check the output of statistical analysis reported in table 1 and S2 . i.e:
Table 1
Is it be possible that CMA2G 0.99±0.16 and CYA 0.81±0.62 are different from 1.02±0.04?
Best regards
Author Response
Thank you for your review and comments.
Point 1: The resolution of figure 1 and 2 must be improved. Concerning figure 1 it is difficult to follow and agree the comments reported in page 3 lines 98-102.
Response 1: Corrected/Improved in Manuscript
Point 2: In table 2 and table S2 must be reported n, the number of independent variables you have. Please, carefully check the output of statistical analysis reported in table 1 and S2 . i.e:
Table 1
Is it be possible that CMA2G 0.99±0.16 and CYA 0.81±0.62 are different from 1.02±0.04?
Response 2: Three replicates per medium were done in the experiment (n=3).
Output of statistical analysis was checked, and 2 mistakes were found and corrected: CMA2G 0.99 ab±0.16; CYA 0.81ab±0.62.

Reviewer 2 Report
The authors evaluated the Production of aflatoxin B1 by Aspergillus parasiticus grown on a novel meat-based media.
The paper is well organized and well written.
Some minor remarka are following.
Line 398. Sánchez-Montero et al., 2019b...should leave.
Please write all references according to the journal's guidelines.
Author Response
Thank you for your review and comments.
Point 1: Line 398. Sánchez-Montero et al., 2019b...should leave.
Response 1: Corrected in the Manuscript.
Point 2: Please write all references according to the journal's guidelines.
Response 2: Corrected in the Manuscript.

Reviewer 3 Report
The work presented for review, although very interesting, in my opinion requires rewording and a different form of presentation. In its current form, it is too basic.
In the introduction, artificially expanded, next to a number of basic data, information about other mycotoxins appears completely unnecessarily.
Acknowledging the CMA2 medium as a novel is controversial. It would be simpler to use peptone. In addition, with such a selection of media, it was possible to use PDB (potato dextrose broth) as a control, next to YES and CYA media, which can also be supplemented.
Setting lag-phase and growth rate is definitely better done in microplate riders (Bioscreen , Tecan).
Conclusions, in general, despite the correct formulation, do not bring anything revealing.
Author Response
Thank you for your review and comments.
Point 1: In the introduction, artificially expanded, next to a number of basic data, information about other mycotoxins appears completely unnecessarily.
Response 1: We are aware that informations about other mycotoxins (mainly ochratoxin A) are presented in the introduction, however we think it is necessary to mention ochratoxin A in dry-cured fermented meat products, as it is the mycotoxin most commonly found in such products, but only few studies have studied the aflatoxins in these products. Therefore, we believe that the proposed content is be suitable for the target researchers.
Point 2: Acknowledging the CMA2 medium as a novel is controversial. It would be simpler to use peptone. In addition, with such a selection of media, it was possible to use PDB (potato dextrose broth) as a control, next to YES and CYA media, which can also be supplemented.
Response 2: YES, MEA and CYA were selected to be used as control media, as they are regularly used for mould growth. Further, it was shown that peptone does not support aflatoxin formation (Yu, 2012; doi: 10.3390/toxins4111024), thus peptone was not chosen as growth medium in the study. The usage of CMA2 as a novel meat-based media (media containing some meat components), especially with ingredients (glucose and yeast extract) that simulate the composition of dry-fermented meat products, appears to be more more suitable for AFB1 production kinetics study in comparison with other meat-based media used in this experiment, based on our results.
Point 3: Setting lag-phase and growth rate is definitely better done in microplate readers (Bioscreen, Tecan).
Response 3: We designed our study with the aim to measure all three parameters (lag-phase, growth rate and AFB1 content) in the same medium in the same Petri dish. We are aware that the lag phase and growth rate measurements can be made in a microtiter plate and with a microtiter plate reader, but we purposely chose a solid culture medium because we wanted to get as close as possible to dried meat products. That is why we also performed the measurements in a solid culture medium and not in a broth in microtitre plates.
Point 4: Conclusions, in general, despite the correct formulation, do not bring anything revealing.
Response 4: Our results provide an overview about quantifying AFB1 production every day during the 15-days incubation period. We think that that the results obtained in this work suggest that A. parasiticus should be considered as a potential risk of AFB1 contamination due to its high capacity of producing AFB1 in meat-based media, which was not generally known before.
On the other hand, global climate shifts, primarily warming, favour mould growth and mycotoxin production, even in European countries with moderate climates and consequently the presence of A. parasiticus on various foods and feeds.

Round 2
Reviewer 3 Report
The revisions have improved the publication and I wish the authors success in their continued research.
Author Response
Thank you for your comments and good wishes.